# Hadamard Test is Sufficient for Efficient Quantum Gradient Estimation with Lie Algebraic Symmetries

**Mohsen Heidari**[*,1],     **Masih Mozakka**[1],     **Wojciech Szpankowski**[2,3]
[1] Indiana University     [2]CSoI, Purdue University     [3]Jagiellonian University

## Abstract

Gradient estimation is a central challenge in training parameterized quantum circuits (PQCs) for hybrid quantum-classical optimization and learning problems. This difficulty arises from several factors, including the exponential dimensionality of the Hilbert spaces and the information loss in quantum measurements. Existing estimators, such as finite difference and the parameter shift rule, often fail to adequately address these challenges for certain classes of PQCs. In this work, we propose a novel gradient estimation framework that leverages the underlying Lie algebraic structure of PQCs, combined with the Hadamard test. By analyzing the differential of the matrix exponential in Lie algebras, we derive an expression for the gradient as a linear combination of expectation values obtained via Hadamard tests. The coefficients in this decomposition depend solely on the circuit's parameterization and can be computed efficiently. Furthermore, these expectation values can be estimated using state-of-the-art shadow tomography techniques. Our approach enables efficient gradient estimation, requiring a number of measurement shots that scales logarithmically with the number of parameters, and with polynomial classical and quantum time. This is an exponential reduction in the measurement cost and a polynomial speed-up in time compared to existing works.

## 1 Introduction

Hybrid quantum-classical strategies have emerged as a leading approach for quantum optimization and learning [BLSF19, CAB+21], and have been extensively studied across a broad range of domains, including optimization [FGG14], quantum chemistry [JEM+19, AWGP21, GEBM19, DAJ+21], and quantum machine learning from classical and quantum data [FN18, SK19, MNKF18, LW18, HCT+19, HPS21, HS24, HS23, HBM+21]. Variational quantum algorithm (VQA) particularly has been a promising paradigm for quantum learning and inference, where a PQC (a.k.a ansatz) is trained in a classical-quantum loop. Gradient-based training methods have gained significant attention in the literature [HN21, SWM+20, SBG+18, FGG14, FN18, SK19, MNKF18] and have demonstrated significant advantages in convergence rates compared to gradient-free methods.

However, estimation of the gradient can be computationally challenging due to several factors including the exponential dimensionality of the Hilbert spaces, the no-cloning, information loss of quantum measurements, and non-commutativity of Hamiltonian terms. Therefore, each gradient estimation can have an exponential sample complexity leading to a high overhead and hence a bottleneck for the scalability of gradient-based VQAs.

Several approaches have been introduced to estimate the gradient [FN18, MNKF18, SWM+20, HGS22, HN21, SBG+18, MKF19, WLW+24, SKP24]; but they often yield suboptimal gradient circuits for certain PQCs. Methods based on finite differences evaluate the objective function in

---

[*]Corresponding author. Email: `mheidar@iu.edu`

the neighborhood of the parameters. They can be applied to general PQCs, but suffer from a slow convergence rate [HN21]. The well-known parameter shift rule (PSR) [SBG$^+$18, MNKF18] relies on the Hadamard test with Pauli operators to estimate the partial derivatives. The Hadamard test is an efficient method that directly measures the partial derivatives and does not have the numerical instability of indirect methods such as finite differences. However, PSR with Hadamard test is restricted to ansätze with two distinct eigenvalues. It can be adapted for more complex circuits via backpropagation, but it comes with high computational costs in terms of gate decomposition. Other existing methods often apply to more general circuits but have high overhead due to the extensive use of the ansatz with repeated measurements, and exponential classical computation [BC21, WIWL22, The23].

Lie algebraic structures in PQCs have been increasingly important in analysis and design of hybrid quantum-classical strategies. Ansätze that have dynamical Lie algebra (DLA) with polynomial dimensionality may not exhibit any Barren plateaus, which are flat regions in the parameter landscape [CSV$^+$21, FHC$^+$23, MBS$^+$18]. Moreover, such Lie algebraic symmetries have be used for classical simulation of quantum models [GLC$^+$23]. In this work, we build upon the Lie algebraic characterizations and we develop an efficient gradient estimation for general circuits based on the Hadamard test followed by post-processing steps. With that, we enable efficient applicability of the Hadamard test to generic PQCs without the need to change the ansatz structure and with low classical overhead.

## 1.1 Summary of The Main Results

We analytically derive an explicit expression for the gradient of generic PQCs in terms of the expectation values of the Hadamard tests corresponding to a certain set of Pauli strings. Then, we develop a gradient estimation method using a series of Hadamard tests at the output of the ansatz followed by classical post-processing techniques including classical shadow tomography (CST) [HKP20]. A generic PQC on $n$ qubits can be represented as $U(\overrightarrow{a}) = e^{iA(\overrightarrow{a})}$, where $A$ is the parameterized Hamiltonian with $\overrightarrow{a} = (a_1, \cdots, a_p)$ as the vector of parameters. Typically, the Hamiltonian is written in term of Pauli strings as $A(\overrightarrow{a}) = \sum_{i=1}^{p} a_i P_i$, where $a_i \in \mathbb{R}$ and $P_i$'s are tensor products of Pauli operators. This formulation includes multi-layered PQCs (e.g., the hardware-efficient ansatz) and appears in a wide range of setups including quantum approximate optimization algorithm (QAOA), many-body quantum systems (e.g., Ising model), and adiabatic evolutions. Typically, the objective is minimizing a loss function $\mathcal{L}(\overrightarrow{a})$ depending on the parameterization of the PQC, the input state, and the measurement observable. Hence, a gradient-based optimization can be used given an estimation of $\nabla \mathcal{L}$.

Our gradient estimation method is based on a binary encoding of Pauli strings to capture the structural properties of their commutation relations. Similar techniques have been used before in the context of the stabilizer formulation in quantum error correction [CRSS96, Got97]. Then, we make a connection between this binary encoding and the differential of the matrix exponential map, studied in Lie algebra [Ros06]. We write the partial derivatives of the PQC as an infinite-length linear combination of expectation values of Hadamard tests for various Pauli strings. We show that when the Pauli strings in $A(\overrightarrow{a})$ are closed under the commutation, the infinite-length linear combination collapses to a finite number of terms that can be computed efficiently using the binary encoding. Such terms are written as the expectation value of a set of observables that can be estimated using CST, as an estimation procedure to estimate several observables with minimal sample complexity [HKP20] growing logarithmically with the number of observables.

When the closedness condition is not directly satisfied, one can consider the sub algebra generated by $P_i$ terms in $A(\overrightarrow{a})$ and apply the proposed gradient estimation method. In that case, the sample complexity and running time depend on the dimensionality of this sub-algebra. Therefore, the proposed estimation is efficient when the dimensionality is polynomial with $n$. The polynomial size assumption is already satisfied for several well-known Hamiltonian models, including variants of 2-local Ising model (e.g., the transverse-field) and Kitaev chain [WKKB23] used to model molecular dynamics. In addition, the polynomial dimensionality is an essential component to avoid the barren plateaus [LCS$^+$22, FHC$^+$23, MBS$^+$18, CSV$^+$21] which are flat regions in the parameters landscape that have exponentially small gradients. This stems from the fact that the variance of the gradient is inversely proportional to the dimension of the DLA [FHC$^+$23].

| | Circuit Changes | Sample Complexity | Running Time* |
|---|---|---|---|
| SPSR[BC21, WIWL22] | $p$ | $O(p)$ | $O(n^{a+b})$ |
| NPSR[The23] | $\tilde{O}(p\|A\|)$ | $\tilde{O}(p\|A\|)$ | $O(n^{a+b})$ |
| $SU(N)$[WLW$^+$24] | $p$ | $O(p)$ | $\exp(\Theta(n))$ |
| [AKH$^+$23] | - | $O(\log^2 p)$ | $p\exp(\tilde{O}(n))$ |
| This work (bounded shadow norm) | **0** | $\mathbf{O}(\log \mathbf{p})$ | $\mathbf{\tilde{O}(n^b + n^{3a})}$ |

Table 1: Rough comparison of various methods for estimating the gradient of a PQC with $p$ parameters and DLA based on Pauli strings with $\text{poly}(n)$ dimensionality. *For presentation convenience of the runtime, it is assumed that $p = \Theta(n^a)$ and that each use of the ansatz takes $\Theta(n^b)$ quantum time, where $a$ and $b$ are arbitrary constants. Also, the detailed dependencies on $\frac{1}{\epsilon}$ and $\|O\|_\infty$ are ignored.

Nevertheless, there still is a curiosity to understand the gradient estimation when DLA dimensionality grows exponentially with $n$. For that, approximation techniques are proposed in the more complete version of the paper [HMS24].

A more specific summary of our contributions is give below:

- Showing the gradient of the loss function can be written as $\nabla\mathcal{L} = \overrightarrow{D}(I - e^V)V^{-1}$, where $\overrightarrow{D}$ is the vector of the expectation values of Hadamard tests for a set of Pauli strings, and $V$ is a matrix constructed based on the parameters $\overrightarrow{a}$ (see Theorem 2).

- An algorithm that estimates $\nabla\mathcal{L}(\overrightarrow{a})$ using $\tilde{O}(p)$ Hadamard tests and $O(p^3 + pn)$ classical time.

- When the shadow norm of the observable is bounded, the gradient can be estimated with $O(\log p)$ copies and $\text{poly}(n)$ time (see Section 3.3).

- A master's theorem for the general case where the Pauli terms are not closed under commutation (see Theorem 4).

## 1.2 Comparison With Related Methods

We consider the three complexity measures: (1) sample complexity, (2) the classical post-processing time, and (3) the number of distinct circuits that need to be evaluated to obtain all the partial derivatives. Although this measure is less restrictive, it is important to ensure it scales polynomially with the number of qubits. Table 1 demonstrates a simplified comparison with existing works for gradient estimation. For a more intuitive comparison, it is assumed that the PQC acts on $n$ qubits with gate complexity $\Theta(n^b)$ and $p = \Theta(n^a)$ parameters, where $a$ and $b$ are arbitrary constants. The table shows that our approach provides an exponential advantage in terms of the copy complexity and a polynomial speed-up in classical running time. Below, we highlight some of the most relevant approaches for comparison to our work.

**Stochastic PSR:** This method is a generalization of PSR [BC21, WIWL22], where each partial derivative is written as an integral, and a Monte Carlo strategy is used to estimate it. David *et al.* [WIWL22] also presented a generalization of the PSR using the Discrete Fourier series. Here, the parameters are jointly shifted depending on the spectrum of the ansatz. This method is efficient when the Hamiltonian $A$ is promised to have equidistant eigenvalues. However, for a generic PQC one first needs to compute the spectral decomposition of $A$ to find the pattern of the parameter shifts. This process in general takes $\exp\{\Theta(n)\}$ classical time as $A$ is an exponentially large matrix.

**Nyquist PSR:** Recently [The23] proposed a shift rule for PQCs where only the parameters are shifted without any other modifications of the ansatz. The method relies on a beautiful connection between the Nyquist-Shannon Sampling theorem and the Fourier series that was observed earlier in [WIWL22, VT18]. The number of unique circuits for this estimation scales with $p$ and the difference between the maximum and minimum eigenvalues of $A$ — a quantity bounded by the operator norm $\|A\|$. As the authors reported, this method has low approximation error when the parameter value is large enough. More precisely, the approximation error is $O(\frac{1}{c^2})$ as long as $\theta = (1 - \Omega(1))c$, where $c$ is the maximum magnitude of a parameter value.

**Lie algebraic:** This is another approach [WLW+24] based on Lie algebra and a nice connection to the geometry of $SU(2^n)$ matrices and the adjoint operator. The gradient is calculated by finding the Jacobian matrix of the matrix representation of PQC. Hence, the running time scales as $p2^{\Theta(n)}$.

**Shadow tomography:** A recent work [AKH+23] proposes a quantum backpropagation method for PQC of the form $U(\overrightarrow{a}) = \prod_{j=1}^{p} e^{ia_j P_j} U_j$, where $U_j$ are fixed unitaries and $P_j$ are fixed Pauli words. Leveraging the shadow tomography of [Aar18], the method achieves a sample complexity scaling as $\log^2 p$, but with the cost of an exponential classical memory requirement of $p2^{\tilde{O}(n)}$.

**Classical simulation:** Existing simulation methods such as g-sim [GLC+23] can efficiently simulate a quantum system and hence compute the gradient, under the assumption that both the Hamiltonian $A(\overrightarrow{a})$ and the observable $O$ have polynomial Lie dimensionality. In contrast, we only require $A(\overrightarrow{a})$ to have polynomial Lie dimensionality.

Assuming both $\rho$ and $O$ are efficiently classically simulatable, our work and that of [GLC+23] have time complexity scaling polynomially with the dimension of the Lie algebra. Our work is complementary to [GLC+23] and offers distinct advantages when $\rho$ and/or $O$ are not classically simulatable. This arises, for example, when $O$ does not lie in a Lie algebra of polynomial dimension, and $\rho$ is a physical quantum state obtained via an external process (e.g., quantum sensing) or generated by a unitary that does not have polynomial Lie dimensionality.

When $\rho$ is stored in a physical quantum state, [GLC+23] relies on computing expectation values of the Lie algebra basis elements ($B_\alpha^{(\lambda)}$ in the reference) under $\rho$. This is only efficient when certain classes of $\rho$ including product states or stabilizer states. In more general setting, such expectation values must be estimated in a quantum computer and the sample complexity of the estimation may not be polynomial in $n$. Furthermore, even when $\rho$ is classically simulatable, we expect significant (possibly exponential) separation from [GLC+23] when the observable does not have polynomial Lie dimensionality while it can be implemented in polynomial quantum time on a quantum device and has bounded shadow norm. For example, in fidelity estimation, $O = |\phi\rangle\langle\phi|$, where $|\phi\rangle = e^{iH}|0\rangle$ and $H$ is a Hamiltonian not admitting a polynomial Lie decomposition. Here, $O$ is a low-rank observable with bounded shadow norm, but exponential Lie dimensionality. Hence, classical simulation runs exponentially in time.

## 2  Preliminaries and Model

### 2.1  General Framework

The objective is to minimize a cost function defined as

$$\mathcal{L}(\overrightarrow{a}) := \text{tr}\{O\,U(\overrightarrow{a})\rho U(\overrightarrow{a})^\dagger\}, \tag{1}$$

where $O$ is a fixed observable, $\rho$ is the initial (mixed) state, and $U(\overrightarrow{a})$ is a parameterized quantum circuit with $\overrightarrow{a} = (a_1, \cdots, a_p)$ as the vector of parameters. As $U$ is unitary, we can always write $U(\overrightarrow{a}) = e^{iA(\overrightarrow{a})}$ for some Hamiltonian matrix $A$. To ensure computational tractability, it is assumed that the number of parameters $p = \text{poly}(n)$, with $n$ being the number of qubits.

Making iterative progress in the direction of the steepest descent is one of the most popular optimization techniques in VQAs, as it has been in classical problems. Ideally, a gradient descent optimizer applies the following update rule at each iteration $t$:

$$\overrightarrow{a}^{(t+1)} = \overrightarrow{a}^{(t)} - \eta_t \nabla\mathcal{L}(\overrightarrow{a}^{(t)}), \tag{2}$$

where $\eta_t \in \mathbb{R}$ is the learning rate at iteration $t$. The above update rule is not realistic as the objective function $\mathcal{L}(\overrightarrow{a})$ is an expectation value, and the characteristics of $\rho$ are either unknown or computationally intractable.

### 2.2  Pauli Group

The Pauli gates are fundamental quantum gates that correspond to rotations around the respective axes of the Bloch sphere and form the basis for many quantum algorithms. Together with the identity,

they are denoted as $\{\sigma^0, \sigma^1, \sigma^2, \sigma^3\}$ with

$$\sigma^0 = I = \begin{pmatrix} 1 & 0 \\ 0 & 1 \end{pmatrix} \qquad \sigma^1 = X = \begin{pmatrix} 0 & 1 \\ 1 & 0 \end{pmatrix}, \quad \sigma^2 = Y = \begin{pmatrix} 0 & -i \\ i & 0 \end{pmatrix}, \quad \sigma^3 = Z = \begin{pmatrix} 1 & 0 \\ 0 & -1 \end{pmatrix}.$$

The single-qubit Pauli group $\mathcal{P}_1$ is the 16 element set $\{c\sigma^s : s = 0, 1, 2, 3, \ c = \pm 1, \pm i\}$. The product of the Pauli operators is governed by the identities $XYZ = iI$ and $X^2 = Y^2 = Z^2 = I$.

For $n$ qubit systems, the Pauli tensor products are denoted as $\sigma^{\mathbf{s}} := \sigma^{s_1} \otimes \sigma^{s_2} \otimes \cdots \otimes \sigma^{s_d}$, for all $\mathbf{s} \in \{0, 1, 2, 3\}^n$. The $n$-qubit Pauli group $\mathcal{P}_n$ is then defined as the group generated by $n$-fold tensor products of the Pauli matrices:

$$\mathcal{P}_n = \{c\sigma^{\mathbf{s}} : \mathbf{s} \in \{0, 1, 2, 3\}^n, \ c = \pm 1, \pm i\}.$$

This group has $4^{d+1}$ elements and spans any operator on the space of $n$ qubits:

**Fact 1.** *Any bounded operator $A$ on $n$ qubits can be uniquely written as $A = \sum_{\mathbf{s} \in \{0,1,2,3\}^n} a_{\mathbf{s}} \, \sigma^{\mathbf{s}}$, where $a_{\mathbf{s}} = \frac{1}{2^n} \operatorname{tr}\left\{ A\sigma^{\mathbf{s}} \right\}$.*

In light of this statement, we can assume that the parameterized circuit of the ansatz is of the form $U(\overrightarrow{a}) = \exp\{iA(\overrightarrow{a})\}$, where $A(\overrightarrow{a}) = \sum_{\mathbf{s} \in \mathcal{S}} a_{\mathbf{s}} \sigma^{\mathbf{s}}$ for some $\mathcal{S} \subseteq \{0, 1, 2, 3\}^n$.

## 2.3 Hadamard Test

Given a unitary $U$, the Hadamard test, which is a special case of the phase estimation, is a quantum circuit that we can use to estimate the real or imaginary value of $\langle \psi | U | \psi \rangle$ for some state $|\psi\rangle$. The circuit consists of two Hadamard gates and a controlled version of $U$.

When the ansatz has a simple form $U(\theta) = e^{i\theta\sigma^{\mathbf{s}}}$ its derivative can be directly estimated via a Hadamard test [MNKF18] giving the following quantity

$$D_{\mathbf{s}} := i \operatorname{tr}\{O[\sigma^{\mathbf{s}}, \rho^{out}]\}, \tag{3}$$

where $\rho^{out} = U(\overrightarrow{a})\rho U(\overrightarrow{a})^\dagger$ is the output of the ansatz on input $\rho$. The above equation is based on the fact that $\frac{\mathrm{d}U}{\mathrm{d}\theta} = i\sigma^{\mathbf{s}}U(\theta)$. Estimating the gradient through the Hadamard test can enhance computing efficiency, and allows for the use of measurement optimization techniques. Moreover, it can be used to compute higher-order partial derivatives used in higher-order optimization algorithms [LDO+24].

## 2.4 Differential of The Matrix Exponential

The matrix exponential is defined as

$$\exp(X) = e^X = \sum_{k=0}^{\infty} \frac{X^k}{k!},$$

where $X$ is a square matrix. Due to non-commutativity of matrix product, the differential of the exponential map has a more complex formula compared to the exponential function. Suppose $X(\tau)$ is a differentiable matrix (linear operator) as a function of the variable $\tau \in \mathbb{R}$. The adjoint map is defined as the mapping $\mathsf{ad}_X(Y) = [X, Y] = XY - YX$ for square $n \times n$ matrixes $X, Y \in \mathsf{GL}(n, \mathbb{C})$. Then, for any $k = 0, 1, ...$, we can define

$$\mathsf{ad}_X^k(Y) = [X, \cdots, [X, Y] \cdots].$$

The exponential map and the adjoint are fundamental concepts in the theory of Lie groups and Lie algebras, describing how a Lie group or Lie algebra acts on its own Lie algebra by conjugation (a standard text book on this topic is [Ros06]). The adjoint operator is connected to the derivative of the matrix exponential.

**Theorem 1** ([Ros06]). *Suppose $X(\tau)$ is a differentiable (linear) operator with respect to a variable $\tau \in \mathbb{R}$. Then, the differential of the matrix exponential is given by*

$$\frac{\mathrm{d}\exp\{X(\tau)\}}{\mathrm{d}\tau} = \exp\{X(\tau)\} \frac{1 - \exp\{-\mathsf{ad}_X\}}{\mathsf{ad}_X} \frac{\mathrm{d}X(\tau)}{\mathrm{d}\tau}. \tag{4}$$

# 3 Main Results

We introduce an approach for estimating the gradient of the loss for a generic ansatz using the Hadamard test followed by classical post-processing. We start with a binary encoding of the Pauli matrices.

## 3.1 Binary Encoding of Pauli Operators

It is well-known that the Pauli group $\mathcal{P}_n$ is *isomorphic* to the *semi direct product* of $\mathbb{Z}_4$ and $\mathbb{Z}_2^{2n}$. In this work, we present an explicit form of such a mapping. This binary representation allows us to write the partial derivatives of $\mathcal{L}(\overrightarrow{a})$ as linear combination of terms related to the Hadamard tests applied to the ansatz output.

Note that the phase scalar $c \in \{\pm 1, \pm i\}$ in $\mathcal{P}_n$ can be written as $i^a$, where $a \in \mathbb{Z}_4$, the modulo-four group. As for the Pauli operators, consider the binary vector group $\mathbb{Z}_2 \times \mathbb{Z}_2 = \{(0|0), (0|1), (1|0), (1|1)\}$ with the element-wise modulo two addition:

$$(a^0|a^1) + (b^0|b^1) = (a^0 \oplus b^0|a^1 \oplus b^1),$$

where $\oplus$ is the binary addition. We use $(\cdot|\cdot)$ to distinguish between the first and the second components of elements of $\mathbb{Z}_2 \times \mathbb{Z}_2$. We associate the identity and each Pauli operator with elements of $\mathbb{Z}_2 \times \mathbb{Z}_2$ as

$$\sigma^0 \to (0|0), \qquad \sigma^1 \to (0|1), \qquad \sigma^2 \to (1|0), \qquad \sigma^3 \to (1|1).$$

Extending to $n$-qubits, $(\mathbb{Z}_2 \times \mathbb{Z}_2)^n$ is defined as the set of all $(\mathbf{a}^0|\mathbf{a}^1)$ for binary strings $\mathbf{a}^0, \mathbf{a}^1 \in \mathbb{Z}_2^n$ with the element-wise addition:

$$(\mathbf{a}^0|\mathbf{a}^1) + (\mathbf{b}^0|\mathbf{b}^1) = (\mathbf{a}^0 \oplus \mathbf{b}^0|\mathbf{a}^1 \oplus \mathbf{b}^1).$$

We sometimes write $\mathbb{Z}_2^{2n}$ to denote this group for more compactly. Any Pauli string $\sigma^{\mathbf{s}}, \mathbf{s} \in \{0, 1, 2, 3\}^n$ is associated with $(\mathbf{s}^0|\mathbf{s}^1)$ where $\mathbf{s}^0 = (s_1^0, \cdots, s_d^0), \mathbf{s}^1 = (s_1^1, \cdots, s_d^1)$ are binary strings. Therefore, we frequently switch between representations of s as a member of $\mathbb{Z}_4^n$ and $(\mathbb{Z}_2 \times \mathbb{Z}_2)^n$.

**Example 1.** *The Pauli string $X \otimes Y \otimes X$ which is associated with $\sigma^{\mathbf{s}}$ with $\mathbf{s} = (1, 2, 1)$ has the following binary encoding:* $((0, 1, 0)|(1, 0, 1))$.

**Definition 1.** *Any element of $\mathcal{P}_n$, written as $i^a\sigma^{\mathbf{s}}$, is represented as $(a, \mathbf{s})$ where $a \in \mathbb{Z}_4$ and $\mathbf{s} \in \mathbb{Z}_2^{2n}$. Such a representation is defined by the mapping $\phi : \mathcal{P}_n \to \mathbb{Z}_4 \times \mathbb{Z}_2^{2n}$, that sends $\phi(i^a\sigma^{\mathbf{s}}) = (a, \mathbf{s})$.*

The above mapping can be used to encode the product of the Pauli matrices.

**Encoding the products of Pauli words.** Inspired by the Levi-Civita symbol, we define a sign function on $i, j \in \mathbb{Z}_4$ (or $\mathbb{Z}_2 \times \mathbb{Z}_2$) as

$$\delta(i, j) := \begin{cases} -1 & \text{if } (i, j) = (1, 3), (2, 1), (3, 2) \\ 1 & \text{if } (i, j) = (1, 2), (2, 3), (3, 1) \\ 0 & \text{otherwise.} \end{cases} \tag{5}$$

For vectors $\mathbf{u}, \mathbf{v}$, define $\delta(\mathbf{u}, \mathbf{v}) = \sum_j \delta(u_j, v_j)$.

**Lemma 1.** *The product of any pair of Pauli strings $\sigma^{\mathbf{s}}, \sigma^{\mathbf{r}}$ equals $\sigma^{\mathbf{s}}\sigma^{\mathbf{r}} = i^{\delta(\mathbf{s}, \mathbf{r})}\sigma^{\mathbf{s} \oplus \mathbf{r}}$.*

*Proof.* It is not difficult to verify the lemma for single qubit case. For general $d > 1$, with the tensor product, we have that

$$\sigma^{\mathbf{s}}\sigma^{\mathbf{r}} = \bigotimes_j \sigma^{s_j}\sigma^{r_j} = \bigotimes_j i^{\delta(s_j, r_j)}\sigma^{s_j \oplus r_j} = i^{\sum_j \delta(s_j, r_j)} \bigotimes_j \sigma^{s_j \oplus r_j} = i^{\sum_j \delta(s_j, r_j)}\sigma^{\mathbf{s} \oplus \mathbf{r}}.$$

$\square$

With this result, and by adding the phase scalars, for any pair $i^a\sigma^{\mathbf{s}}$ and $i^b\sigma^{\mathbf{r}}$ from the Pauli group $\mathcal{P}_n$, the product is characterized by the $\phi$ map in Definition 1 as

$$(i^a\sigma^{\mathbf{s}})(i^b\sigma^{\mathbf{r}}) = \phi^{-1}((\text{mod}_4(a + b + \delta(\mathbf{s}, \mathbf{r})), \mathbf{s} \oplus \mathbf{r})).$$

**Example 2.** *Consider* $P_1 = iX \otimes Y \otimes X$ *and* $P_2 = Z \otimes X \otimes Y$ *that are encoded to* $(1, ((010)|(101)))$ *and* $(0, ((101)|(110)))$, *respectively. Then,* $P_1 P_2$ *is associated the binary encoding* $(0, ((111)|(011)))$ *which represents* $Y \otimes Z \otimes Z$.

This binary representation is used to characterize the commutation relations between Pauli strings.

**Lemma 2.** *The commutator of any pair of Pauli strings* $\sigma^{\mathbf{s}}, \sigma^{\mathbf{r}}$ *is given by* $[\sigma^{\mathbf{s}}, \sigma^{\mathbf{r}}] = 2i^{\delta(\mathbf{s}, \mathbf{r})} \sigma^{\mathbf{s} \oplus \mathbf{r}}$.

### 3.2 Gradient Estimation

Recall that the objective function $\mathcal{L}(\overrightarrow{a})$ in (1) and the expectation values $D_{\mathbf{s}}$ in (3) can be estimated using Hadamard test. In what follows, we show how the gradient of $\mathcal{L}$ can be evaluated in terms of $D_{\mathbf{s}}$ quantities. Recall the Hamiltonian Pauli decomposition: $A(\overrightarrow{a}) = \sum_{\mathbf{s} \in \mathcal{S}} a_{\mathbf{s}} \sigma^{\mathbf{s}}$. We first assume that the set of Pauli strings $\sigma^{\mathbf{s}}, \mathbf{s} \in \mathcal{S}$ appearing in this decomposition is closed under the commutation, that is for any pair $\mathbf{s}, \mathbf{t} \in \mathcal{S}$ the commutator $[\sigma^{\mathbf{s}}, \sigma^{\mathbf{r}}]$ also appears in the above decomposition. The more general case where the Pauli terms are not closed under the commutation is also considered in a more complete version of the paper [HMS24].

We consider a geometric representation of the gradient in terms of $D_{\mathbf{s}}$ terms. Let $\mathcal{S} = \{\mathbf{s}_1, \cdots, \mathbf{s}_p\}$, where $p$ is the number of parameters. Let $\mathbf{e}_1 = (1, 0, \cdots, 0)^T, \cdots, \mathbf{e}_p = (0, \cdots, 0, 1)^T$ be the canonical basis vectors in $\mathbb{R}^p$. We associate each $\mathbf{s}_j$ with $\mathbf{e}_j$ which is also denoted by $\mathbf{e}_{(\mathbf{s}_j)}$. Now consider the vector of the expectation terms $\overrightarrow{D} = (D_{\mathbf{s}_1}, \cdots, D_{\mathbf{s}_p})$ as a row vector in $\mathbb{R}^p$. Then, the gradient is expressed as a vector derived from $\overrightarrow{D}$.

**Theorem 2.** *For the ansatz* $U(\overrightarrow{a}) = \exp\{iA(\overrightarrow{a})\}$, *with* $A(\overrightarrow{a}) = \sum_{\mathbf{s} \in \mathcal{S}} a_{\mathbf{s}} \sigma^{\mathbf{s}}$, *where* $\mathcal{S}$ *is closed under commutation, the gradient satisfies* $\nabla \mathcal{L} = \overrightarrow{D}(I - e^V)V^{-1}$, *where* $V$ *is the* $p \times p$ *matrix with the* $i$*th column given by*

$$\mathbf{v}_j := 2i \sum_{\mathbf{s} \in \mathcal{S}} a_{\mathbf{s}} i^{\delta(\mathbf{s}, \mathbf{s}_j)} \, \mathbf{e}_{(\mathbf{s} \oplus \mathbf{s}_j)}. \tag{6}$$

*Moreover there is an algorithm (Algorithm 1) that computes* $\nabla \mathcal{L}$ *in* $O(p^3 + pn)$ *time with* $O(p)$ *use of the Hadamard tests.*

The proof outline is given in Section 4.2. This representation is the key to the efficient computation of the gradient from the estimation of $D_{\mathbf{s}}, \mathbf{s} \in \mathcal{S}$, summarized as Algorithm 1. The runtime of the algorithm is $O(p^3 + pn)$ because each column $\mathbf{v}_i$ is computed in $O(pn)$ time and, and the matrix $B$ can be computed in $O(p^3)$ time with a proper matrix exponentiation algorithm.

---

**Algorithm 1** Gradient Estimation

**Input:** $\mathcal{S}$

1: **procedure** GRADIENT ESTIMATION
2:    Estimate $D_{\mathbf{s}}, \mathbf{s} \in \mathcal{S}$ with Hadamard tests.
3:    Compute the matrix $V$ where the column $i$ is computed as in (6).
4:    Compute the matrix $B = (I - e^{-V})V^{-1}$, where $V^{-1}$ is the generalized inverse of $V$.
5: **Return** $\widehat{\nabla \mathcal{L}} = \overrightarrow{D} B$.

---

A simple example of a Hamiltonian closed under commutation is $A = a_1 X^{\otimes n} + a_2 Y^{\otimes n} + a_3 Z^{\otimes n}$. The gradient in this case can be computed by a $3 \times 3$ matrix and 3 Hadamard tests, irrespective of the dimensionality of the quantum system. Another class is $K$-junta Hamiltonians, that acting non-trivially on $k$ qubits. In that case, all the Pauli strings $\sigma^{\mathbf{s}}, \mathbf{s} \in \mathcal{S}$, will be of the form $\sigma_k^{\mathbf{s}} \otimes I_{n-k}$, the set $\mathcal{S}$ will be closed under commutation.

**Remark 1.** *When the closedness condition is not directly satisfied, one can apply Theorem 2 to the DLA generated by the terms in* $A(\overrightarrow{a})$. *When the dimensionality of the DLA is* $m$, *Algorithm 1 computes* $\nabla \mathcal{L}$ *in* $O(m^3 + dm)$ *time with* $O(m)$ *use of the Hadamard tests. Hence, Algorithm 1 is efficient when the DLA has dimensionality polynomial in* $n$.

**Example 3.** *The following Hamiltonian has a DLA with dimensionality* $O(n^2)$:

$$H = \sum_{i=1}^{n} Z_i Z_{i+1} + X_i,$$

where $X_i, Z_i$ are the corresponding Pauli operators. In this case, applying Theorem 2 to the DLA generated by $H$, Algorithm 1 computes $\nabla\mathcal{L}$ in $O(n^6)$ time with $O(n^2)$ use of the Hadamard tests.

As a sanity check, Appendix A present an explicit derivation of the gradient in the single qubit case.

### 3.3 Efficient Estimation with Classical Shadow Tomography

Turning the gradient estimation to a series of Hadamard tests has another benefit that can further reduce the number of shots to $O(\log p)$. This can be done using shadow tomography [Aar18, HKP20, HBM+21, CGY24, KGKB25]. The components of the gradient $\nabla\mathcal{L}$ correspond to $p$ observables that only depend on $\rho^{out}$ without any reconfigurations. In that case, having several copies of $\rho^{out}$ we can efficiently estimate all the components of the gradient. Based on Theorem 2, the observables are $p$ different Hadamard tests denoted by $H_{\mathbf{s}_j}, j \in [p]$ for measuring $D_{\mathbf{s}_j}$. The following lemma gives an explicit characterization of these observables.

**Lemma 3.** *It holds that $D_{\mathbf{s}} = \mathrm{tr}\{H_{\mathbf{s}}\rho^{out}\}$, where $\rho^{out}$ is the ansatz output state and $H_{\mathbf{s}} = R^{\dagger}_{\mathbf{s},-}OR_{\mathbf{s},-} - R^{\dagger}_{\mathbf{s},+}OR_{\mathbf{s},+}$, with $R_{\mathbf{s},\pm} = \exp\{-i\frac{\pm\pi}{4}\sigma^{\mathbf{s}}\}$ for Pauli string $\sigma^{\mathbf{s}}$.*

*Proof.* Recall that $D_{\mathbf{s}} = i\,\mathrm{tr}\{O[\sigma^{\mathbf{s}}, \rho^{out}]\}$. Moreover, from [MNKF18] the following property holds for any operator $B$ and Pauli string $\sigma^{\mathbf{s}}$:

$$[\sigma^{\mathbf{s}}, B] = i\Big(R_{\mathbf{s}}(\frac{\pi}{2})BR_{\mathbf{s}}(\frac{\pi}{2})^{\dagger} - R_{\mathbf{s}}(-\frac{\pi}{2})BR_{\mathbf{s}}(-\frac{\pi}{2})^{\dagger}\Big),$$

where $R_{\mathbf{s}}(\theta) := e^{-i\frac{\theta}{2}\sigma^{\mathbf{s}}}$. With this observation, we obtain

$$D_{\mathbf{s}} = -\mathrm{tr}\Big\{O\Big(R_{\mathbf{s}}(\frac{\pi}{2})\rho^{out}R_{\mathbf{s}}(\frac{\pi}{2})^{\dagger} - R_{\mathbf{s}}(-\frac{\pi}{2})\rho^{out}R_{\mathbf{s}}(-\frac{\pi}{2})^{\dagger}\Big)\Big\}$$

$$= -\mathrm{tr}\Big\{R_{\mathbf{s}}(\frac{\pi}{2})^{\dagger}OR_{\mathbf{s}}(\frac{\pi}{2})\rho^{out}\Big\} + \mathrm{tr}\Big\{R_{\mathbf{s}}(-\frac{\pi}{2})^{\dagger}OR_{\mathbf{s}}(-\frac{\pi}{2})\rho^{out}\Big\},$$

where we used the cyclic property of the trace. The last equation gives the expression for $H_{\mathbf{s}}$. $\square$

In order to estimate the gradient, one needs to estimate all the expectation values $o_{\mathbf{s}}^{\pm} := \mathrm{tr}\Big\{R^{\dagger}_{\mathbf{s},\pm}OR_{\mathbf{s},\pm}\rho^{out}\Big\}$ for $\mathbf{s} \in \mathcal{S}$. We use CST to exponentially reduce the measurement shots for the gradient estimation.

**Theorem 3** ([HKP20]). *Given an observable $O$, state $\rho^{out}$, and a set $\mathcal{S} \subseteq \{0, 1, 2, 3\}^n$, the expectation values $o_{\mathbf{s}}^{\pm} := \mathrm{tr}\Big\{R^{\dagger}_{\mathbf{s},\pm}OR_{\mathbf{s},\pm}\rho^{out}\Big\}$ for $\mathbf{s} \in \mathcal{S}$ can be estimated with $\epsilon$ additive error using*

$$N = O\bigg(\frac{1}{\epsilon^2}\log|\mathcal{S}|\max_{\mathbf{s}\in\mathcal{S}}\Big\|R^{\dagger}_{\mathbf{s},\pm}OR_{\mathbf{s},\pm}\Big\|^2_{shadow}\bigg)$$

*copies of $\rho^{out}$, where $\|\cdot\|_{shadow}$ is the shadow norm.*

The shadow norm (Definition 2 in Appendix B) is closely related to the variance of the observable and the set of the unitary transformation used for taking the classical shadows. For random Clifford measurements, it is bounded by the Hilbert-Schmidt norm; whereas for random Pauli measurements, it is bounded by $4^k$, where $k$ is the the locality of the observable, not the actual number of qubits [HKP20]. For completeness, a brief summary of CST is provided in Appendix B.

## 4 Proofs Overview

### 4.1 Master's Theorem

We prove a master's theorem for the partial derivatives of an ansatz with generic subset $\mathcal{S}$ that is not necessarily closed under the commutations. This result can be used as is for generic PQCs and as a step to prove our main results.

**Theorem 4.** *For the ansatz $U(\overrightarrow{a}) = \exp\{iA(\overrightarrow{a})\}$ with $A(\overrightarrow{a}) = \sum_{\mathbf{s} \in \mathcal{S}} a_{\mathbf{s}} \sigma^{\mathbf{s}}$ for some generic subset $\mathcal{S} \subseteq \{0, 1, 2, 3\}^n$, the partial derivative with respect to $a_{\mathbf{r}}$ is given by*

$$\frac{\partial \mathcal{L}(\overrightarrow{a})}{\partial a_{\mathbf{r}}} = \sum_{k=0}^{\infty} \frac{(2i)^k}{(k+1)!} \sum_{\mathbf{s}_1 \in \mathcal{S}} \cdots \sum_{\mathbf{s}_k \in \mathcal{S}} \left( \prod_{j=1}^{k} a_{\mathbf{s}_j} i^{\delta(\mathbf{s}_j, \mathbf{s}_1 \oplus \cdots \oplus \mathbf{s}_{j-1} \oplus \mathbf{r})} \right) D_{\mathbf{s}_1 \oplus \cdots \oplus \mathbf{s}_k \oplus \mathbf{r}}. \tag{7}$$

*Proof outline.* Noting that $\rho^{out} := U\rho U^\dagger$, the partial derivative of the loss with respect to $a_{\mathbf{s}}$ is $\frac{\partial \mathcal{L}}{\partial a_{\mathbf{s}}} = \text{tr}\left\{ O \frac{\partial}{\partial a_{\mathbf{s}}} \rho^{out} \right\}$, where

$$\frac{\partial \rho^{out}}{\partial a_{\mathbf{s}}} = \frac{\partial U}{\partial a_{\mathbf{s}}} \left( \rho U^\dagger \right) + (U\rho) \frac{\partial U^\dagger}{\partial a_{\mathbf{s}}}, \tag{8}$$

and we used the fact that $\rho^{out}$ is Fréchet differentiable with respect to $a_{\mathbf{s}}$. Based on Theorem 1, we prove a slightly different expression of the differential of the matrix exponential.

**Lemma 4.** *Suppose $U = \exp\{iH(\tau)\}$, where $H(\tau)$ is differentiable. Then, $\frac{dU}{d\tau} = -\frac{1-\exp\{i\text{ad}_H\}}{\text{ad}_H}(\frac{dH(\tau)}{d\tau})U$.*

*Proof.* We can write $U = (e^{-iH(\tau)})^\dagger$. Therefore, given that $\frac{dX^\dagger}{d\tau} = (\frac{dX}{d\tau})^\dagger$, from (4) we can write

$$\frac{dU}{d\tau} = \left( \frac{de^{-iH(\tau)}}{d\tau} \right)^\dagger = \left( \exp\{-iH(\tau)\} \frac{1 - \exp\{-\text{ad}_{-iH}\}}{\text{ad}_{-iH}} \frac{d(-iH(\tau))}{d\tau} \right)^\dagger$$

$$= \left( \frac{1 - \exp\{-\text{ad}_{-iH}\}}{\text{ad}_{-iH}} \frac{d(-iH(\tau))}{d\tau} \right)^\dagger U.$$

Note that for any $H \in \mathfrak{g}$ we have the following equality by its convergent power series:

$$\frac{1 - \exp\{-\text{ad}_H\}}{\text{ad}_H} = \sum_{k=0}^{\infty} \frac{(-1)^k}{(k+1)!} (\text{ad}_H)^k. \tag{9}$$

Therefore, replacing $H$ with $-iH$ in this equation, the derivative equals to the following

$$\frac{dU}{d\tau} = \left( -i \sum_{k=0}^{\infty} \frac{(i)^k}{(k+1)!} (\text{ad}_H)^k (\frac{dH}{d\tau}) \right)^\dagger U = i \sum_{k=0}^{\infty} \frac{(-i)^k}{(k+1)!} \left( (\text{ad}_H)^k (\frac{dH}{d\tau}) \right)^\dagger U.$$

Note that for any $X, Y \in \mathfrak{g}$, $(\text{ad}_X(Y))^\dagger = -\text{ad}_X^\dagger(Y^\dagger)$. Therefore,

$$\frac{dU}{d\tau} = i \sum_{k=0}^{\infty} \frac{(i)^k}{(k+1)!} (\text{ad}_H)^k (\frac{dH}{d\tau}) U = -\frac{1 - \exp\{+i\text{ad}_H\}}{\text{ad}_H} (\frac{dH}{d\tau}) U.$$

$$\square$$

From this lemma, we obtain the first part of (8):

$$\frac{\partial \rho^{out}}{\partial a_{\mathbf{s}}} = -\frac{1 - \exp\{i\text{ad}_A\}}{\text{ad}_A} (\frac{\partial A}{\partial a_{\mathbf{s}}}) (U\rho U^\dagger) + (U\rho) \frac{\partial U^\dagger}{\partial a_{\mathbf{s}}}.$$

Next, from Theorem 1 and (4), the partial derivative of $U^\dagger$ can be written as $\frac{\partial U^\dagger}{\partial a_{\mathbf{s}}} = U^\dagger \frac{1 - \exp\{+i\text{ad}_A\}}{\text{ad}_A} (\frac{\partial A}{\partial a_{\mathbf{s}}})$, where we used the fact that $A$ is Hermitian. Therefore, the partial derivative of $\rho^{out}$ equals to

$$\frac{\partial \rho^{out}}{\partial a_{\mathbf{s}}} = -\frac{1 - \exp\{i\text{ad}_A\}}{\text{ad}_A} (\frac{\partial A}{\partial a_{\mathbf{s}}}) \rho^{out} + \rho^{out} \frac{1 - \exp\{i\text{ad}_A\}}{\text{ad}_A} (\frac{\partial A}{\partial a_{\mathbf{s}}}).$$

By simplifying the terms in the right-hand side, the partial derivative can be written as the commutator:

$$\frac{\partial \mathcal{L}}{\partial a_{\mathbf{s}}} = \text{tr} \left\{ O \frac{\partial}{\partial a_{\mathbf{s}}} \rho^{out} \right\} = \text{tr} \left\{ O \left[ \rho^{out}, \frac{1 - \exp\{i\text{ad}_A\}}{\text{ad}_A} (\frac{\partial A}{\partial a_{\mathbf{s}}}) \right] \right\}. \tag{10}$$

Next, based on the Taylor expansion of the matrix exponential and from the fact that $\frac{\partial A}{\partial a_{\mathbf{s}}} = \sigma^{\mathbf{s}}$, the above quantity decomposes as

$$\frac{\partial \mathcal{L}}{\partial a_{\mathbf{s}}} = -i \sum_{k=0}^{\infty} \frac{(i)^k}{(k+1)!} \operatorname{tr}\{O[\rho^{out}, (\mathsf{ad}_A)^k(\sigma^{\mathbf{s}})]\}. \tag{11}$$

Then, building on the binary encoding of Pauli operators, and Lemma 2, we can write the adjoint operator as below.

**Lemma 5.** *For any $A, B$, $\mathsf{ad}_A(B) = 2\sum_{\mathbf{r},\mathbf{s}} a_{\mathbf{s}} b_{\mathbf{r}} i^{\delta(\mathbf{s},\mathbf{r})} \sigma^{\mathbf{s}\oplus\mathbf{r}}$, where $a_{\mathbf{s}} := \frac{1}{2^n}\operatorname{tr}\{A\sigma^{\mathbf{s}}\}$ and $b_{\mathbf{r}} := \frac{1}{2^n}\operatorname{tr}\{B\sigma^{\mathbf{r}}\}$ are the Pauli coefficients of $A$ and $B$, respectively.*

Therefore, omitting the details, we can write the series decomposition of $\mathsf{ad}_A^k(\sigma^{\mathbf{r}})$, appearing in (11), and as the partial derivative in the statement of Theorem 4. $\qquad\square$

### 4.2 Proof of Theorem 2

*Proof outline.* The proof of this results is based on the master theorem. Since the Pauli terms $\sigma^{\mathbf{s}}$, $\mathbf{s} \in \mathcal{S}$ are closed under commutation, then per Lemma 2, $\mathcal{S}$ is closed under the "$\oplus$" operation, that is $\mathbf{s} \oplus \mathbf{t} \in \mathcal{S}$ for any pair $\mathbf{s}, \mathbf{t} \in \mathcal{S}$. Hence, each term $\mathbf{s}_1 \oplus \cdots \oplus \mathbf{s}_k \oplus \mathbf{r}$ in (7) in Theorem 4 remains in $\mathcal{S}$. As a result the infinite-length sum in Theorem 4 reduces to a linear combination with finite terms as $\frac{\partial \mathcal{L}}{\partial a_{\mathbf{r}}} = \sum_{\mathbf{s}\in\mathcal{S}} g_{\mathbf{s}}(\mathbf{r}) D_{\mathbf{s}}$, where $g_{\mathbf{s}}(\mathbf{r}) \in \mathbb{R}$ are some coefficients. This means that the partial derivative is always a linear combination of $D_{\mathbf{s}}, \mathbf{s} \in \mathcal{S}$ terms. However, the challenge is in computing the coefficients $g_{\mathbf{s}}(\mathbf{r})$ that are coming from an infinite-length sum. We present a method to address this issue.

Recall that $V$ is the $p \times p$ matrix with the $j$th column given by $\mathbf{v}_j := 2i\sum_{\mathbf{s}\in\mathcal{S}} a_{\mathbf{s}} i^{\delta(\mathbf{s},\mathbf{s}_j)} \mathbf{e}_{(\mathbf{s}\oplus\mathbf{s}_j)}$, By an induction argument, and from the definition of the matrix exponential, it is not difficult to check that

$$(I - e^V)V^{-1}\mathbf{e}_{(\mathbf{r})} = \sum_{k=0}^{\infty} \frac{(2i)^k}{(k+1)!} \sum_{\mathbf{s}_1\in\mathcal{S}} \cdots \sum_{\mathbf{s}_k\in\mathcal{S}} \Big(\prod_{j=1}^{k} a_{\mathbf{s}_j} i^{\delta(\mathbf{s}_j, \mathbf{s}_1\oplus\cdots\oplus\mathbf{s}_{j-1}\oplus\mathbf{r})}\Big) e_{\mathbf{s}_1\oplus\cdots\oplus\mathbf{s}_k\oplus\mathbf{r}}.$$

If one replaces $\mathbf{e}_{\mathbf{s}}$ with $D_{\mathbf{s}}$ for any $\mathbf{s}$, the we obtain (7) for the partial derivative of $\mathcal{L}$. Omitting some details, the gradient is then calculated by $\nabla\mathcal{L} = \overrightarrow{D}(I - e^V)V^{-1}$. $\qquad\square$

## 5 Discussion and Conclusion

This paper provides a framework to estimate the gradient of generic PQCs via Hadamard tests for Pauli operators followed by classical post-processing. It is shown that the proposed approach is polynomial in classical and quantum resources when the DLA of the associated Hamiltonian of the PQC has a dimensionality polynomial in the number of qubits. Moreover, this method does not change the ansatz structure and can be used to reduce the measurement shot complexity to scale logarithmically with the number of parameters. The results would be beneficial in various optimization or learning quantum algorithms that rely on the estimation of the gradient.

One limitation of this work is when the Hamiltonian has exponentially many Pauli terms. In that case, we can only approximate the gradient by truncation of the nested summations in Theorem 4 to a fixed number of terms. However, this will be a biased approximation. As future work, one can extend the proposed framework to the estimation of higher-order derivatives. Deriving lower bounds on the classical and quantum resources needed to estimate the gradient or higher-order derivatives is another important direction.

## Acknowledgments and Disclosure of Funding

This work is partially supported by the NSF Center for Science of Information (CSoI) Grant CCF-0939370, and also by NSF Grants CCF-2006440 and and CCF-2211423.

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

# A Derivation of The Gradient for a Single Qubit PQC

Consider a general single-qubit unitary of the form

$$U(\overrightarrow{a}) = \exp\{i(a_1\sigma^1 + a_2\sigma^2 + a_3\sigma^3)\}.$$

Let $O$ be a generic observable and consider the associated loss $\mathcal{L}(\overrightarrow{a})$ as in (1). As an illustrative example, we examine the partial derivative of $\mathcal{L}$ with respect to $a_1$, evaluated at the point $a_1 = 0$ and $a_3 = 0$. In the following we compute this derivative using two approaches: first, by applying Theorem 4, and second, through direct calculation.

**Closed-form expression based on Theorem 4.** In the context of Theorem 4, let $D_j$ be the result of the Hadamard test with Pauli $\sigma^j$, where $j = 1, 2, 3$. Then, (7) in Theorem 4 simplifies to the following:

$$\frac{\partial \mathcal{L}}{\partial a_1}(\overrightarrow{a} = (0, a_2, 0)) = \sum_{k=0}^{\infty} \frac{(2i)^k}{(k+1)!} \left( \prod_{j=1}^{k} a_2 i^{\delta(2,(\overbrace{2 \oplus \cdots \oplus 2}^{j-1 \text{ times}} \oplus 1))} \right) D_{(\underbrace{2 \oplus \cdots \oplus 2}_{k \text{ times}} \oplus 1)},$$

where we used the fact that only terms with $\mathbf{s}_j = 2$ are surviving. Because $a_1 = a_3 = 0$. Note that for even $j$ we have $\underbrace{2 \oplus \cdots \oplus 2}_{j-1 \text{ times}} \oplus 1 = 3$, and for odd $j$ it is equal to $0 \oplus 1 = 1$. Therefore,

$$\delta(2, (\underbrace{2 \oplus \cdots \oplus 2}_{j-1 \text{ times}} \oplus 1)) = \begin{cases} \delta(2,3) = 1 & \text{even } k \\ \delta(2,1) = -1 & \text{odd } k \end{cases}$$

where we used (5). Plugging it in the first equation, we have

$$\prod_{j=1}^{k} a_2 \left( i^{\delta(2,(\overbrace{2 \oplus \cdots \oplus 2}^{k-1 \text{ times}} \oplus 1))} \right) = a_2^k(-i) \times i \times (-i) \times \cdots = \begin{cases} (a_2 i)^k (-1)^{k/2} & \text{even } k \\ (a_2 i)^k (-1)^{(k+1)/2} & \text{odd } k. \end{cases}$$

As a result, the partial derivative simplifies to

$$\frac{\partial \mathcal{L}}{\partial a_1}(\overrightarrow{a} = (0, a_2, 0)) = \sum_{p=0}^{\infty} \frac{(-2)^{2p}}{(2p+1)!} a_2^{2p} (-1)^p D_1 + \sum_{q=0}^{\infty} \frac{(-2)^{2q+1}}{(2q+2)!} a_2^{2q+1} (-1)^{q+1} D_3.$$

Hence, one needs to measure $D_1$ and $D_3$ to compute the above partial derivative. Next, by simplifying the summations, it is not difficult to show that

$$\frac{\partial \mathcal{L}}{\partial a_1}(\overrightarrow{a} = (0, a_2, 0)) = \frac{-1}{2a_2} \left( \sum_p \frac{(-2a_2)^{2p+1}}{(2p+1)!} (-1)^p \right) D_1 + \frac{-1}{2a_2} \left( \sum_q \frac{(-2a_2)^{2q+2}}{(2q+2)!} (-1)^{q+1} \right) D_3$$

$$= \frac{-1}{2a_2} \left( \sin(-2a_2) D_1 + (\cos(-2a_2) - 1) D_3 \right)$$

$$= \frac{1}{2a_2} \left( \sin(2a_2) D_1 + (1 - \cos(2a_2)) D_3 \right).$$

Notice the presence of $D_3$ which relates to the Pauli $\sigma^3$ and not appear in the ansatz expression. One can verify that this is indeed equal to the analytic gradient of this ansatz.

**Direct derivation.** Note that from (8) the partial derivative of the objective function can be written as

$$\frac{\partial \mathcal{L}}{\partial a_{\mathbf{s}}} = \text{tr}\left\{ O\left( \frac{\partial U}{\partial a_{\mathbf{s}}}(\rho U^\dagger) + (U\rho)\frac{\partial U^\dagger}{\partial a_{\mathbf{s}}} \right) \right\},$$

Given that $\frac{\partial U^\dagger}{\partial a_\mathbf{s}} = (\frac{\partial U}{\partial a_\mathbf{s}})^\dagger$. Then, by denoting $\tilde{U} = \frac{\partial U}{\partial a_\mathbf{s}}$ we have that

$$\frac{\partial \mathcal{L}}{\partial a_\mathbf{s}} = \mathrm{tr}\Big\{ O\tilde{U}\rho U^\dagger + U\rho\tilde{U}^\dagger \Big\}.$$

Next, as $UU^\dagger = I$, we have

$$\frac{\partial \mathcal{L}}{\partial a_\mathbf{s}} = \mathrm{tr}\left\{ O\left( \tilde{U}U^\dagger(U\rho U^\dagger) + (U\rho U^\dagger)U\tilde{U}^\dagger \right) \right\}$$

$$= \mathrm{tr}\left\{ O\left( \tilde{U}U^\dagger\rho^{out} + \rho^{out}(\tilde{U}U^\dagger)^\dagger \right) \right\},$$

where $\rho^{out}$ is the ansatz output. Note that the single qubit ansatz can also be written as

$$U(\overrightarrow{a}) = I\cos\theta + i\left( \sum_{s\in\{0,1,2,3\}} \hat{a}_s\sigma^s \right)\sin\theta, \tag{12}$$

where $\theta = \sqrt{\sum a_s^2}$ is a normalizing parameter and $\hat{a}_s = \frac{a_s}{\theta}$. Now, we can differentiate $U$ with respect to a single parameter $a_s$ appearing in the sum:

$$\frac{\partial U(\overrightarrow{a})}{\partial a_s} = (-\frac{a_s}{\theta}\sin\theta)I + i\left( \sum_{s'\neq s} \frac{-a_s a_{s'}}{\theta^3}\sigma^{s'} + \frac{\theta^2 - a_s^2}{\theta^3}\sigma^s \right)\sin\theta$$

$$+ i\left( \frac{a_s\cos\theta}{\theta}\sum_{s'}\hat{a}_{s'}\sigma^{s'} \right). \tag{13}$$

Therefore,

$$\frac{\partial U(\overrightarrow{a})}{\partial a_1}\bigg|_{a_1=a_3=0} = i\sigma^1\frac{\sin a_2}{a_2}.$$

Using (12) to find $U^\dagger$, we have that

$$\tilde{U}U^\dagger = i\frac{\sin a_2}{a_2}\left( \cos a_2\sigma^1 + \sin a_2\sigma^3 \right).$$

Which when plugged into the derivative expression gives

$$\frac{\partial L}{\partial a_1}\bigg|_{a_1=0} = \frac{\sin a_2}{a_2}\left( \cos a_2 D_1 + \sin a_2 D_3 \right) = \frac{\sin 2a_2}{2a_2}D_1 + \frac{(1-\cos 2a_2)}{2a_2}D_3.$$

This is identical to the expression based on Theorem 4.

## B   Classical Shadow Tomography

For completeness, in this section we briefly describe the classical shadow tomography procedure. For more details see [HKP20]. Classical shadow tomography is a technique used in quantum computing to efficiently learn properties of a quantum state using only a few measurements. It was introduced to extract useful information from quantum states without requiring a full quantum state tomography, which is costly in terms of the number of measurements and computational resources.

More precisely, let $O_j, j \in [M]$ be a set of observables. The goal is to estimate the expectation value of these observable for measuring an unknown state $\rho$ in a Hilbert space $\mathcal{H}$.

**Theorem 5** ([HKP20]). *Suppose the observables $O_j, j \in M$ are traceless, then the expectation values $\mathrm{tr}\{O_j\rho\}, j \in [M]$ can be approximated up to an additive error $\epsilon$ with probability $(1-\delta)$ with*

$$O\left( \frac{1}{\epsilon^2}\log\frac{M}{\delta}\max_j\|O\|_{shadow}^2 \right)$$

*copies of $\rho$.*

The shadow norm is a measure that resembles the variance in the worst case state, defined in the following. We first describe the steps in this procedure for a generic state $\rho$ in a Hilbert space $\mathcal{H}$.

First, generate a unitary operator $U$ randomly from a class of choices $\mathcal{U}$ to be determined. Apply $U$ on the input state resulting in the state $U^\dagger \rho U$. Measure the resulted state in the canonical basis $|j\rangle, j \in [\dim_{\mathcal{H}}]$. From Born's rule the probability of getting the output $j$ is $p_j = \langle j|U^\dagger \rho U|j\rangle$. Given an outcome $j$, define $\omega_j = U|j\rangle\langle j|U^\dagger$. The expectation $\mathbb{E}_{\sim(J,U)}[\omega_J]$ over the measurement randomness ($p_j$) and the choice of $U$ equals to $\mathcal{M}[\rho]$, where $\mathcal{M}$ is a mapping defined as

$$\mathcal{M}[O] := \mathbb{E}_U\Big[\sum_{j\in[\dim_{\mathcal{H}}]} \langle j|U^\dagger OU|j\rangle \; U|j\rangle\langle j|U^\dagger\Big], \tag{14}$$

for any operator $O$ on $\mathcal{H}$. Observe that $\mathcal{M}$ is a linear mapping on $\mathcal{B}(\mathcal{H})$ and hence has an inverse denoted by $\mathcal{M}^{-1}$. We note that $\mathcal{M}^{-1}$ is the shadow channel $\mathcal{M}^{-1}$ introduced in [HKP20]. We apply $\mathcal{M}^{-1}$ on $\omega_j$ resulting in the so called shadow

$$\hat{\rho} := \mathcal{M}^{-1}\big[U|j\rangle\langle j|U^\dagger\big]. \tag{15}$$

Note that $\hat{\rho}$ is a classical matrix and hence can be copied several times. Moreover, $\hat{\rho}$ is not a valid density operator as it is not necessarily a positive semi-definite matrix. However, it is an unbiased estimate of the original state.

When $\mathcal{U}$ is tomographically complete, the classical shadow $\hat{\rho}$ is unbiased, that is $\mathbb{E}_{U,J}[\hat{\rho}] = \rho$.

**Definition 2.** *The shadow norm of any operator $O$ with respect to a set $\mathcal{U}$ of tomographically complete unitaries is defined as*

$$\|O\|_{\text{shadow}} := \max_{\sigma\in\mathcal{D}[H]} \Big(\mathbb{E}_{U\sim\mathcal{U}} \sum_{j\in[dim_{\mathcal{H}}]}\langle j|U^\dagger\sigma U|j\rangle \; \langle j|U\mathcal{M}^{-1}[O]U^\dagger|j\rangle^2\Big)^{1/2}.$$

## B.1   CST with Pauli Measurements

**Shadow tomography with Pauli measurements.** Suppose the observable $O_j$ act non trivially on at most $k$ qubits. For that $U$ in CST is the tensor product of randomly chosen Pauli operators:

$$U = U_1 \otimes \cdots \otimes U_n \in CL(2)^{\otimes n},$$

where each $U_j$ is chosen randomly and uniformly from the Clifford group $CL(2)$. In this case, $\mathcal{U} = CL(2)^{\otimes n}$. Moreover, the shadow matrix is computed as

$$\hat{\rho} := \bigotimes_{j=1}^{n} \Big(3U_j^\dagger \big|\hat{b}_j\big\rangle\big\langle\hat{b}_j\big| U_j - I\Big).$$

In that case, the shadow norm is bounded by the locality of the observables as

$$\|O_j\|_{shadow} \le 2^k\|O_j\|_\infty$$

As a result the sample complexity of CST is given by

$$O\Big(\frac{4^k}{\epsilon^2}\log m \max_j \|O_j\|_\infty^2\Big).$$

The CST algorithm runs in $\tilde{O}\big(2^{\Theta(k)}m\log m\big)$ classical time.

## B.2   CST with Clifford Measurements

The *Clifford group* is a set of unitary operations that map Pauli operators to other Pauli operators under conjugation. For a system of $n$ qubits, the Clifford group $CL(2^n)$ consists of unitaries $U$ such that for any Pauli string $P$:

$$UPU^\dagger = P',$$

where $P'$ is another Pauli operator.

Clifford circuits are particularly useful because they can be efficiently simulated classically, and their structure allows for easy manipulation of Pauli observables, making them useful for shadow

tomography. In that case, $U$ is chosen randomly from the Clifford group. The shadow norm is bounded as

$$\|O_j\|_{shadow} \leq \sqrt{3\operatorname{tr}\{O^2\}},$$

when $\operatorname{tr}\{O^2\} < \infty$. The shadow matrix is given by

$$\hat{\rho} = (2^n + 1)U^\dagger \left|\hat{b}\right\rangle\!\left\langle\hat{b}\right| U - I.$$

As a result the sample complexity of CST is bounded as

$$n = O\!\left(\frac{1}{\epsilon^2}\log m \max_j \operatorname{tr}\{O_j^2\}\right),$$

and the CST algorithm runs $\tilde{O}\!\left(2^{\Theta(n)}m\log m\right)$ classical time.

