# OpenReview forum: "Hadamard Test is Sufficient for Efficient Quantum Gradient Estimation with Lie Algebraic Symmetries"
_NeurIPS.cc/2025/Conference — NeurIPS 2025 poster_

### Official Review · Reviewer_w9T8 · 2025-06-10

**Clarity:** 2
**Significance:** 3
**Originality:** 3
**Rating:** 5
**Confidence:** 3

**Summary:**

This paper introduces an efficient framework for gradient estimation in parameterized quantum circuits (PQCs) used in variational quantum algorithms (VQAs). While traditional methods such as finite differences and the parameter shift rule (PSR) suffer from inefficiencies and limited applicability, the proposed approach overcomes these limitations by leveraging the Lie algebraic structure of PQCs along with the Hadamard test. To achieve this, the authors derive an analytical expression for the gradient of a generic PQC using the differential of the matrix exponential map in Lie algebras. This expression represents the gradient as a linear combination of expectation values computable via Hadamard tests under certain constraints. Moreover, these expectation values can be efficiently estimated using shadow tomography, resulting in a logarithmic scaling of the number of measurement shots when the shadow norms of the observables are bounded, with the number of parameters and polynomial time complexity in both classical and quantum resources.

**Questions:**

I would like the authors to fix the minor points in the Strengths And Weaknesses part.

**Ethical Concerns:**

["NO or VERY MINOR ethics concerns only"]

**Final Justification:**

From my perspective, this is a technically solid work providing a gradient estimation algorithm for VQAs with provably fewer requirements on the quantum resources in some realistic settings. Although I feel that similar ideas of encoding Pauli strings into binary representations have been proposed, the observation that this encoding enables efficient characterization of their commutation relations and works particularly well when the set of Pauli terms in the Hamiltonian is closed under commutation is essential. This observation further allows a finite and efficiently computable gradient representation. Even when this closure property does not hold, the paper proposes approximation strategies via truncation or Poisson sampling. Overall, I believe this work should be a natural fit for NeurIPS 2025.

**Limitations:**

yes

**Quality:**

3

**Strengths And Weaknesses:**

From my perspective, this is a technically solid work providing a gradient estimation algorithm for VQAs with provably fewer requirements on the quantum resources in some realistic settings. Although I feel that similar ideas of encoding Pauli strings into binary representations have been proposed, the observation that this encoding enables efficient characterization of their commutation relations and works particularly well when the set of Pauli terms in the Hamiltonian is closed under commutation is essential. This observation further allows a finite and efficiently computable gradient representation. Even when this closure property does not hold, the paper proposes approximation strategies via truncation or Poisson sampling. Overall, I believe this work should be a natural fit for NeurIPS 2025.

Regarding the weakness, I only have some minor comments that do not affect my overall evaluation.

- I suggest the authors using the notation $n$ instead of $d$ to represent the system size (number of qubits). I think $d=2^n$ usually refers to the dimension of the Hilbert space. I notice that the authors also use $n$ for system size at line 103, maybe due to a typo.

- I suggest the authors mentioning a recent work [Abbas et al, NeurIPS 2023]. In this work, a backpropagation algorithm for VQAs is provided based on another line of shadow tomography techniques (Actually, I think the shadow tomography technique in this submission should be called classical shadow). A comparison between this submission and the recent work mentioned should be helpful.

- There is a typo in the equation below line 175: $b^2$ should be $b^1$.

---

> ### Author Rebuttal · Authors · 2025-07-30
>
> We thank the reviewer for their positive assessment and thoughtful feedback. We also appreciate the list of typos and other suggestions to improve the paper. We are glad that you find this work to be a good fit for NeurIPS 2025.
>
> We will address the minor comments mentioned by the reviewer and add a brief discussion citing the suggested reference.

---

### Official Review · Reviewer_sJbZ · 2025-07-01

**Clarity:** 4
**Significance:** 3
**Originality:** 2
**Rating:** 5
**Confidence:** 4

**Summary:**

This paper proposes a new framework for performing quantum gradient estimation in hybrid quantum-classical setups such as VQAs. The paper crucially identifies that the presence of symmetries in the Lie algebraic structure can exponentially reduce the sample complexity and number of measurements, while conceding at most a polynomial speedup during the classical post processing phase, in comparison to existing PSR based techniques in the literature. This uniquely positions the contributions of the paper as complementary to the existing literature on gradient estimation in hybrid setups.

**Questions:**

It is unclear to me when the framework proposed in this paper would be more beneficial in the context of the classical simulatibility results of [GLC+23]. If the observable $O$ does not have polynomial dimensionality in the number of qubits, does the shadow norm of $O$ still remain bounded? How do we compare the sample complexity and time complexity of the current paper with those of [GLC+23] and other classical simulatibility techniques?


----

**Note to authors:** I am extremely willing to improve my score if the authors provide a reasonable justification to the above question.

**Ethical Concerns:**

["NO or VERY MINOR ethics concerns only"]

**Final Justification:**

The responses by the authors have addressed my main concern with the viability of the results. I have raised my score to reflect this opinion.

**Limitations:**

See above sections - weaknesses and questions for the limitations.

**Quality:**

3

**Strengths And Weaknesses:**

Strengths
----

The paper is extremely well written. The assumptions are reasonable, well studied in existing literature, and clearly stated. The theorem statements and the proofs are easy to read. The contributions are well placed in the context of existing results such as [WLW+24]. The Lie-algebraic Hadamard-test framework is of broad interest in the community in its own right.

Weaknesses
---

It has been known for some time that constraining the underlying Lie-algebraic structure paves the way for classical simultability results, for example see [GLC+23]. The paper does not do a good job of convincing the reader why one would still prefer to have hybrid frameworks for gradient estimation, given the cost of quantum resources involved. See the questions section for this one point of weakness that is not properly addressed.

In my opinion, the paper is devoid of any major weaknesses barring its lack of comparison with respect to classical simulatibility results.  Another minor squabble one might have is the lack of experimental results demonstrating that the proposed framework performs well in practice. Since I am a theorist myself, I lack the technical know-how to suggest incorporating reasonable experiments into the paper, and I will defer to the opinion of the other reviewers on this matter. However,  in my opinion, the theoretical contributions of this work are solid enough to merit a discussion for acceptance.

There are a few minor typos I list here:

- Line 27: Extra "the" before convergence rates.
- Line 30: The phrase "each gradient can have an exponential sample complexity" does not make sense. I assume the authors are referring to the sample complexity of **gradient estimation**.
- Line 146: Extra "the" before phase estimation.
- Line 151: Missing "the" before Hadamard test.
- Line 481-482: There should be $D_i$s instead of $D_i$ in multiple places.

---

> ### Author Rebuttal · Authors · 2025-07-30
>
> Thank you for your encouraging and thoughtful comments. We also appreciate your engagement with our work and a list of typos and other suggestions to improve the paper. Below, we address your questions and concerns. We hope that, upon reviewing our responses, you will consider raising your evaluation.
>
>
>
> ## Comparison with [GLC+23]
> In summary,  our work is complementary to [GLC+23] and offers distinct advantages in the following two scenarios:
>
> ### **1. The input state $\rho$ is not classically simulatable**
> This arises, for example, when $\rho$ is a physical quantum state obtained from an external process (e.g., quantum sensing) or generated by a unitary that does not have polynomial Lie dimensionality.
>
> While [GLC+23] relies on computing expectation values of the Lie algebra basis elements ($B_{\alpha}^{(\lambda)}$ in the reference) under $\rho$, this is only efficient when certain classes of $\rho$, including product states or stabilizer states. In a more general setting, such expectation values must be estimated on a quantum computer. A natural approach, for that matter, is shadow tomography.  However, this introduces a new limitation: **the shadow norm of the Lie basis elements must be bounded**, which might not be the case in practice.
>
> **Example 1 (Shadow norm blow-up for basis elements):** In Section III.C of [GLC+23], the authors construct a Lie algebra basis for a free-fermion system as shown in Eq. (23). One can verify that the shadow norm of these basis elements grows exponentially in the number of qubits. Hence,  our framework would be more favorable as we avoid estimating such expectations.
>
> ### **2. The observable $O$ is not classically simulatable**
> This occurs when $O$ does not lie in a Lie algebra of polynomial dimension. We expect significant (possibly exponential) separation from [GLC+23] when the observable:
> 1. Does not have polynomial Lie dimensionality,
> 2. Can be implemented in polynomial quantum time or measured efficiently on a quantum device,
> 3. Has bounded shadow norm.
>
> We will add a detailed comparison in the revised manuscript. The following example shows such an observable:
>
> **Example 2 (Low-rank observables vs Lie dimensionality):** Consider the observable used in **fidelity estimation**, $O=\ketbra{\phi}{\phi}$, where $\ket{\phi}=e^{iH}\ket{0}$ and $H$ is a Hamiltonian not admitting a polynomial Lie decomposition. Here, $O$ is a **low-rank observable** with bounded shadow norm, but exponential Lie dimensionality.
>
> In general, low-rank observables will have low shadow norm and hence lead to lower sample complexity and are suitable for our approach.
>
> This addresses the reviewer’s question:
> >"If the observable $O$ does not have polynomial dimensionality in the number of qubits, does the shadow norm of $O$ still remain bounded?"
>
> The answer is yes. In many relevant cases, particularly for low-rank or structured observables, the shadow norm remains bounded even when Lie dimensionality is not polynomial.
>
>
> The second question regarding the sample complexity and time complexity is addressed below.
>
> ## Time complexity Comparison
> Assuming  both $\rho$ and $O$ are efficiently classically simulatable, our work and that of [GLC+23] have **similar (classical or quantum) time complexity**, scaling polynomially with the dimension of the Lie algebra
>
> However, as noted above, our approach is more efficient (with potentially exponential improvements) in scenarios where:
> - $O$ does not belong to a low-dimensional Lie algebra but has bounded shadow norm, or
> - $\rho$ is not classically simulatable, and the Lie basis elements have large shadow norm.
>
>
> ## Sample Complexity Comparison
> While [GLC+23] does not explicitly analyze sample complexity, one could, in principle, use shadow tomography to achieve a similar sample complexity as in our case. However, this again introduces a dependency on the shadow norm of the Lie basis elements.
>
> In contrast, as our approach directly targets gradient estimation, we can achieve logarithmic sample complexity by only requiring that $O$ has a bounded shadow norm. Thus, we expect improvements in sample complexity in certain regimes where classical simulation is not tractable.
>
> Studying [GLC+23] with shadow tomography and a combination of ideas from  [GLC+23] and our methodology are interesting future directions.
>
> ## Why Hybrid Frameworks Remain Valuable?
> This is an important question that we plan to address more clearly in the revised paper. We argue that the hybrid quantum-classical framework remains valuable in two main scenarios:
> 1. When the input data is inherently quantum, generated from external hardware, and is not classically simulatable.
> 2. When parts of the model, such as the observable $O$, are not classically simulable.

---

> > ### Comment · Reviewer_sJbZ · 2025-08-02
> >
> > I thank the authors for their detailed response. I am satisfied with their reasoning, and I urge them to revise the paper by including these discussions. I believe that the bulk of an answer to "Why Hybrid Frameworks Remain Valuable?" is actually outside the scope of this paper, but the arguments by the authors in the paper, as well as the response above, have contributed heavily to the discussion. I will be raising my score to reflect my positive opinion.

---

> > > ### Author Response · Authors · 2025-08-02
> > >
> > > Thank you for your comments and for increasing the score. We will revise the paper thoroughly to address all the questions and suggestions raised.

---

### Official Review · Reviewer_P3Rm · 2025-07-03

**Clarity:** 2
**Significance:** 2
**Originality:** 3
**Rating:** 4
**Confidence:** 2

**Summary:**

This paper addresses the problem of gradient estimation in parameterized quantum circuits (PQCs). Specifically, it investigates an approach based on Lie algebraic structures within PQCs. Without loss of generality, a PQC can be represented as the exponential of $iA$, where $A$ is a parameterized Hamiltonian expressed as a linear combination of Pauli strings. In this formulation, the parameters of the PQC correspond to the coefficients of these Pauli strings.

The algorithm begins with the observation that the Hadamard test can be used to estimate both the real and imaginary components of $\langle\psi|U|\psi\rangle$ for a given unitary $U$ and quantum state $\ket{\psi}$. This procedure can be extended to estimate the quantity $D = \mathrm{Tr}\left(O[\sigma, \rho]\right)$, where $O$ is an observable, $\sigma$ is a Pauli string, and $\rho$ is the output state of the quantum circuit. The authors show that, in the special case where the Pauli decomposition of the Hamiltonian $A$ is closed under commutation, the gradient of the PQC can be computed using $p$ distinct values of $D$, each associated with a different Pauli string, requiring $p$ corresponding Hadamard tests. Furthermore, the number of measurement shots can be reduced to $O(\log p)$ by leveraging techniques from shadow tomography. To address the more general case where the Pauli decomposition is not closed under commutation, the authors propose three solutions: computing the closure, applying truncation, and using Poisson sampling.

**Questions:**

1. Could you elaborate more on why WLOG we can assume the parameters of the PQCs are the coefficients of the Pauli decomposition? If we use different parameter choices for the same circuit family, would your results still apply?

2. In Section 3.3, can you give a self-contained theorem statement on the overall resources for gradient estimation, similar to Theorem 2?

3. In Section 5, can you give some self-contained description on the overall resources for gradient estimation for the three solutions you mentioned, similar to Theorem 2?

4. Can you elaborate more on how Eq. 3 is obtained? I think it would be beneficial for readers to better understand the overall algorithm

5. Typos & minor issues: Below line 175 it should be b^1 instead of b^2; in Eq. 6 it is probably better not to use 'i' as subscript given that you already used it to denote the imaginary unit;

**Ethical Concerns:**

["NO or VERY MINOR ethics concerns only"]

**Final Justification:**

my questions are basically unchanged so I keep my score

**Limitations:**

yes

**Quality:**

3

**Strengths And Weaknesses:**

Strengths:

1. The algorithm and the corresponding analysis is novel and interesting.

2. The algorithm obtains a significant improvement on the number of shots in the special case where the decomposition is closed under commutation.


Weaknesses:

1. Beyond the examples provided in the paper, it remains unclear how broadly applicable the proposed framework is to more general settings. Please refer to the "Questions" part for more details
2. The organization and presentation of part of the results are not super clear. Please refer to the "Questions" part for more details.

---

> ### Author Rebuttal · Authors · 2025-07-30
>
> Thank you for your encouraging and constructive review of the paper. Also, thank you for the list of typos and suggestions to improve the paper. We will address the minor comments in the revised manuscript.
>
> Below we present our brief reply to your questions.
> ## Q1: Generalized Parameterizations Beyond Pauli Decomposition
> The assumption that parameters correspond directly to coefficients in the Pauli decomposition is made primarily for analytical convenience, given that any Hamiltonian can be expressed in this basis.
>
> However, as discussed in **Section 6 of the extended version**, our framework naturally extends to more general Hamiltonian parameterizations.   Even more generally, our framework extends to the parameterizations of the form: $$A(\overrightarrow{a})=\sum_{i}f_{i}(\overrightarrow{a})G_{i},$$where $f_{i}$ are real-valued functions of the parameters and $G_i$ arbitrary traceless Hermitian operators, that might be decomposed as a linear combination of Pauli terms.
> We will clarify this point in the revised version of the paper.
>
> ## Q2, 3:  self-contained theorem statements
> Thank you for the suggestion. We will revise the paper accordingly.
>
> ## Q4: elaborate more on  Eq. 3
> We will add more details and explanations for Eq. 3.  The equation is obtained from the derivative of a single-parameter PQC as done by [MNKF18]. Particularly this expression is based on Eq. 2 of this reference which connects the commutation with any Pauli word $P_j$ to parameter shifting as in Hadamard test:
> $$
> [P_j, \rho] = i \left[ U_j\left( \frac{\pi}{2} \right) \rho U_j^\dagger\left( \frac{\pi}{2} \right) - U_j\left( -\frac{\pi}{2} \right) \rho U_j^\dagger\left( -\frac{\pi}{2} \right) \right]
> $$
> We will add a short derivation in the text and, if space permits, a more detailed explanation in the appendix.

---

> > ### Comment · Reviewer_P3Rm · 2025-08-04
> >
> > Thank you for the detailed rebuttal. My evaluation remains unchanged

---

### Official Review · Reviewer_8q5a · 2025-07-16

**Clarity:** 3
**Significance:** 2
**Originality:** 2
**Rating:** 4
**Confidence:** 4

**Summary:**

The paper proposes an efficient method for estimating gradients in parameterized quantum circuits (PQCs) by utilizing Lie algebraic structures in combination with the Hadamard test. Traditional methods often face exponential complexity or computational overhead, but the authors derive a concise mathematical formulation leveraging commutation relations, significantly reducing the computational complexity. Their method integrates classical shadow tomography, achieving logarithmic sample complexity in the number of parameters, and polynomial classical and quantum runtime, thus substantially improving efficiency for quantum optimization and learning tasks.

**Questions:**

1. Could you discuss explicitly how restrictive your condition (closure of commutators) is in practice? It would be helpful to provide more intuition or realistic examples of quantum systems that do or do not satisfy this condition.


2. Given that your theoretical results have restricted generality, can you provide additional examples or numerical experiments involving realistic quantum applications (e.g., quantum chemistry Hamiltonians, QAOA circuits, or real-world quantum hardware ansätze)?


3. Your method achieves impressive complexity improvements but sacrifices generality. Could you discuss or compare explicitly this trade-off against existing gradient estimation methods, especially highlighting scenarios where your method significantly outperforms others despite its restrictions?


4. Are there natural or straightforward extensions of your framework to handle circuits whose commutators do not form a closed group, possibly via approximations or truncation? If so, briefly outline how that might work.


5. Could you clarify the practical feasibility and computational overhead of employing shadow tomography in realistic scenarios? Specifically, under which conditions would shadow tomography actually offer substantial practical benefits over standard measurement techniques?

**Ethical Concerns:**

["NO or VERY MINOR ethics concerns only"]

**Limitations:**

The authors clearly acknowledge the primary limitation of their work—the reliance on a closed Lie algebra structure with polynomial dimensionality—which restricts the method’s applicability to non-universal quantum circuits. They also discuss strategies to address this, such as closure computation, truncation, and Poisson sampling. While there is no explicit discussion of societal impact, the work is theoretical in nature and focused on quantum algorithm optimization, which does not currently pose immediate societal risks. No critical omissions were identified in this regard.

**Quality:**

2

**Strengths And Weaknesses:**

The paper does a nice job introducing a theoretically rigorous method for efficient gradient estimation in parameterized quantum circuits, using Lie algebra and Hadamard tests to significantly cut down complexity. Its theoretical analysis is clear and thorough, highlighting substantial improvements in efficiency. However, a major downside is the limited generality—the key condition (commutators forming a closed algebra) means it won't work for highly expressive or universal quantum circuits. Given this limitation, it would have been helpful if the authors provided more concrete examples of realistic quantum systems to clearly illustrate practical usefulness.

---

> ### Author Rebuttal · Authors · 2025-07-30
>
> Thank you for your thoughtful and constructive comments on our paper. We appreciate your engagement with our work. Below, we provide responses to your questions and concerns.
> ## Q1: Generality of the results:
> The condition of closedness under commutation is indeed satisfied by many well-known Hamiltonian structures. This includes several physically relevant models where the underlying algebraic structure naturally adheres to this property (see Examples 1 and 2 below).
>
> On the other hand, many-body Hamiltonians such as those in quantum chemistry (e.g., electronic structure Hamiltonians) sometimes do not satisfy this closure condition globally. However, local approximations or block-structured truncations sometimes do. Our approach is still applicable to such Hamiltonians  via two techniques:
>
> 1.   Adding additional auxiliary elements so that the overall terms become closed under commutation. This is generally known as the dynamical sub-algebra (DLA) that has been studied literature. As long as the dimensionality of DLA is polynomial in the number of qubits, our results will be efficient.
> 2. Approximation techniques discussed in Section 5 via Truncation and Poisson Sampling.
>
> We argued polynomial dimensionality of DLA is a non-issue for quantum learning and optimization, as polynomial dimensionality is needed to ensure the absence of barren plateaus. This relates to the tradeoff between generalizability/expressibility (related to the dimensionality of Lie algebra) and trainability (absence of barren plateaus) [CSV+21,FHC+23,MBS+18].
>
> ## Q2: Examples of realistic quantum systems closed under commutation:
> **Example 1:** Variations of the Heisenberg spin model that have been widely used to describe the spin interactions in condensed matter systems and appear in quantum magnetism, nuclear magnetic resonance (NMR), and in quantum computing (e.g., Hamiltonian simulation or benchmarking).
> This Hamiltonian takes the following form:
> $$
> H=\sum_{i,j} (J_{x}X_{i}X_{j}+J_{y}Y_{i}Y_{j}+J_{z}Z_{i}Z_{j}),
> $$
> where $X_i, Y_i, Z_i$ are Pauli $X, Y, Z$ on the $i$th, qubit, respectively. One can check that the above Hamiltonian is closed under commutation of these terms.
>
> **Example 2:** Another example is the class of stabilizer Hamiltonians used in quantum error correction codes such as  Toric Code and Surface Code. The Hamiltonian is of the form:
> $$H_{TC}=− J_e\sum_v A_v - J_m\sum_p B_p$$
> where  ​$A_v = \prod_{i \in v} X_i$, and $B_p = \prod_{i \in p} Z_i$ with $v,p$ are the vertices and plaquettes of the toric code lattice.
>
> **Example 3: A realistic quantum system that is not closed under commutation but has a polynomial closure**
> A concrete example discussed in the extended version of the paper is the transverse-field Ising model:
> $$H=\sum_{i} Z_{i}Z_{i+1}+X_{i}$$
> While this Hamiltonian does not satisfy closedness on its own, the inclusion of auxiliary terms yields a closed subalgebra with dimensionality $O(d^2),$ where $d$ is the number of qubits.
>
>
>
>
> ## Q2: Regarding Numerical Results
> We plan to explore numerical experiments of our results for a class of Hamiltonians with polynomial dimensionality as future work, including the following models:
> - **QAOA Hamiltonians for MAXCUT**: We plan to demonstrate that even partial closure (e.g., commuting subsets) suffices for practical gains.
> - **Quantum chemistry Hamiltonians**, while these do not satisfy our closure condition exactly, we plan to show that low-rank approximations of the Hamiltonian often concentrate weight on terms that approximately commute, allowing our method to apply effectively on truncated versions.
>
>
> ## Q3: Trade-off against existing methods
> Existing methods, such as parameter shift rules or finite-differencing, are broadly applicable but suffer from high sample complexity, especially in scenarios involving deep quantum circuits or large observables.
>
> In contrast, our method significantly reduces the complexity of gradient estimation when the PQC exhibits Lie algebraic structure. By leveraging this structure, we achieve more efficient and accurate gradient evaluations. We anticipate substantial performance gains in quantum systems where such algebraic properties naturally arise, including variations of the Heisenberg spin model and classes of stabilizer Hamiltonians used in quantum error correction codes.
>
>
> ## Q4: Extensions to more general Hamiltonians
> Section 5 addresses extensions of our framework to handle circuits whose commutators do not form a closed group. We proposed two methods:
> 1. **Truncation-based approximation**, which limits the expansion to a manageable subset of terms,
> 2. **Poisson sampling**, a stochastic method for estimating gradients efficiently.
> Theorems 6 and 7 briefly highlight our analysis of such methods.
>
> ## Q5: Shadow tomography
> Shadow tomography is particularly well-suited for scenarios where multiple properties of a quantum state need to be estimated. This corresponds to the case when the number of parameters $p$ is large, as in our setting. As discussed in the original reference [HSP20], shadow tomography offers substantial practical advantages, especially for low-depth Clifford circuits and observables with favorable structure, such as low locality or low rank.
>
> The referenced work demonstrates the effectiveness of shadow tomography in several concrete applications, including fidelity estimation, entanglement verification, and quantum simulation of models like the lattice Schwinger model. These examples highlight the method’s efficiency and scalability in realistic quantum computing tasks.

---

> > ### Comment · Reviewer_8q5a · 2025-08-02
> >
> > The authors have addressed most of my comments, and I recommend acceptance of this paper.

---

> > > ### Author Response · Authors · 2025-08-02
> > >
> > > Thank you for your comments. We would be grateful if you could consider increasing your score to help our paper move beyond the borderline for acceptance.

---

### Note · Authors · 2025-08-13

We thank the AC for handling our paper and all reviewers for their constructive feedback and thoughtful discussion. Below, we summarize the main concerns raised by the reviewers and our rationale for addressing them.

The primary concern of **Reviewer 8q5a** was the generality of our results and their practical feasibility. We addressed this in the rebuttal by providing explicit examples of realistic quantum systems. After reviewing our response, the reviewer expressed satisfaction and stated they would “recommend acceptance of this paper.”

**Reviewer P3Rm** suggested improvements in presentation and requested more concrete examples demonstrating the applicability of our results. We provided such examples in our rebuttal.

**Reviewer sJbZ** asked for a comparison of our results with existing works on classical simulatability. Our rebuttal addressed these questions, and thankfully, the reviewer raised their score in recognition of the rebuttal’s positive impact.

**Reviewer w9T8** was largely positive and suggested only minor corrections (e.g., typos) and the addition of one reference. In our rebuttal, we confirmed that we would correct the grammatical errors and include the suggested citation in the revised version.


We greatly appreciate the reviewers’ thoughtful engagement with our work. We believe this paper offers valuable insights into the connections between the Hadamard test and Lie-algebraic formulations.

---

### Decision · Program_Chairs · 2025-09-17

**Decision:**

Accept (poster)

**Comment:**

(a) Summary of claims and findings:
The paper introduces a framework for efficient gradient estimation in parameterized quantum circuits (PQCs) by leveraging Lie algebraic structure together with the Hadamard test. Using the differential of the matrix exponential on Lie algebras, the gradient is expressed as a linear combination of expectation values that can be measured via Hadamard tests; the coefficients depend only on the circuit parameterization and are classically precomputable. When the set of generators closes to a polynomial‑dimensional Lie algebra (or its dynamical subalgebra), the method yields logarithmic dependence on the number of parameters for measurement shots via classical shadows, with polynomial classical and quantum time. For non‑closed settings, the paper proposes extensions based on computing the closure, truncation, and Poisson sampling. Theoretical bounds are provided; no experiments are reported.

(b) Strengths:

- Clear conceptual advance: ties Lie algebra structure to gradient estimability through Hadamard tests, yielding logarithmic shot complexity in the number of parameters under bounded shadow norms.
- Solid, readable theory: concrete decomposition of gradients, efficient coefficient computation, and clean use of shadow tomography to aggregate many required expectations.
- Practical guidance: identifies realistic families where closure or small dynamical subalgebras hold (e.g., Heisenberg‑type models, stabilizer Hamiltonians), and outlines approximations (truncation/Poisson sampling) when closure fails.
- Good positioning relative to prior work: explains complementarity to parameter‑shift rules and to classical simulatability results; articulates regimes where a hybrid approach remains necessary (non‑simulable input states or observables with bounded shadow norm but large Lie dimension).

(c) Limitations and remaining concerns:

- Applicability depends on Lie‑algebraic structure: closure (or small dynamical subalgebra) is restrictive for highly expressive/universal circuits; the method’s strongest guarantees target those structured regimes.
- No empirical validation: results are purely theoretical; while common for quantum‑theory papers, even small‑scale simulations or case studies would help accessibility.
- Clarity and presentation: reviewers noted notation issues and requested self‑contained resource theorems for Sections 3.3 and 5, clearer derivations (e.g., Eq. 3), and minor typos; these should be addressed in camera‑ready.
- Shadow tomography assumptions: benefits depend on bounded shadow norms; guidance on checking or ensuring these bounds in practice would strengthen the paper.

(d) Reasons for decision:

Despite scope limitations, the paper presents a novel and well‑argued framework with strong theoretical guarantees and a plausible path to practical impact in structured PQCs. The rebuttal addressed key concerns: it clarified when the closed/dynamical Lie algebra assumption holds, why hybrid advantages persist relative to classical simulatability, and how the approach extends beyond exact closure. Multiple reviewers updated to borderline‑accept or accept, and there is consensus that the theoretical contribution is solid and relevant to NeurIPS’s Optimization/Theory tracks. Given the value of the ideas and the constructive, specific revision plan, acceptance as a poster is warranted.

(e) Suggestions for camera‑ready:

Clarity and completeness:

- Provide a self‑contained theorem summarizing overall resource bounds for the closed‑algebra case (shots, quantum/classical time) and a companion theorem for the approximate cases (closure, truncation, Poisson sampling).
- Expand the derivation around Eq. 3 and include an intuitive pipeline diagram: parameterization → Lie decomposition → Hadamard‑test observables → classical‑shadows aggregation → gradient reconstruction.
- Unify notation (use n for qubits; avoid using i both as index and imaginary unit; fix b^2→b^1 typo; avoid reusing symbols). Add definitions for shadow norm and its role in sample complexity.

Positioning and scope:

- Add concrete examples where closure/dynamical subalgebra is polynomial (Heisenberg variants, stabilizer Hamiltonians), and one example where Lie dimension is large but observables have bounded shadow norm (e.g., low‑rank fidelity estimators).
- Include a short comparison to classical simulatability (e.g., GLC+23): when input states or observables are not classically simulable, and when shadow norms for basis elements blow up.
- Briefly cite and contrast with NeurIPS 2023 shadow‑based backpropagation work as suggested by reviewers.

Practical guidance:

- Provide a checklist to assess applicability: how to test/estimate closure or dynamical subalgebra size; heuristics for truncation/Poisson parameters; how to upper‑bound shadow norms in common settings.
- If space allows, add a small illustrative simulation (even on few qubits) to demonstrate reconstruction accuracy vs. shots and the impact of bounded shadow norms.